# Environmental Risk of Leptospirosis in Animals: The Case of the Republic of Sakha (Yakutia), Russian Federation

**DOI:** 10.3390/pathogens9060504

**Published:** 2020-06-23

**Authors:** Olga I. Zakharova, Fedor I. Korennoy, Nadezhda N. Toropova, Olga A. Burova, Andrey A. Blokhin

**Affiliations:** 1Federal Research Center for Virology and Microbiology, Nizhny Novgorod Research Veterinary Institute-Branch of Federal Research Center for Virology and Microbiology, 603950 Nizhny Novgorod, Russia; 79875409639@yandex.ru (N.N.T.); burovaolga@list.ru (O.A.B.); and.bloxin2010@yandex.ru (A.A.B.); 2Federal Center for Animal Health (FGBI ARRIAH), 600901 Vladimir, Russia

**Keywords:** leptospirosis, MaxEnt, risk map, Republic of Sakha (Yakutia), climate change

## Abstract

Leptospirosis is a zoonotic natural focal disease caused by the pathogenic bacteria Leptospira. Its spread is related to certain ecological factors. The aim of the current research was to assess potential exposure to the infection as a function of environmental determinants in the Republic of Sakha (Yakutia), Russian Federation. We applied environmental niche modeling using leptospirosis cases in livestock and wild animals in 1995–2019 with regard to a set of landscape, climatic, and socioeconomic variables, both for the current climate and for the projected climate for 2041–2060. The MaxEnt model performed well (AUC = 0.930), with the mean temperature of the warmest quarter, mean diurnal range, land cover type, and altitude being the most contributing variables. Consequent zoning based on the proportion of high-risk cells within each administrative unit suggested that five out of the 36 districts of the Republic are at high risk in the current climate conditions, with three more districts expected to demonstrate a high risk by 2060. This study presents the first-ever attempt at leptospirosis ecological modeling in Russia. Its results correspond well to the findings of other authors and underline the importance of considering ecological factors when conducting a leptospirosis risk assessment.

## 1. Introduction

Leptospirosis is an infectious natural focal disease caused by the pathogenic bacteria *Leptospira* [1,2]. Currently, it is one of the most widely spread and neglected spirochetal zoonoses [3,4,5,6,7,8,9,10,11]. The pathogenic *Leptospira*, etiological agents of leptospirosis, cause at least 1,000,000 cases of the disease in humans and about 60,000 deaths annually [4]. Leptospirosis can be transmitted directly or indirectly from animals to humans, while human-to-human and human-to-animal transmission occur very rarely [5,6,7,8,9,10,11,12,13,14,15,16]. Many animal species, both domestic and wild, are affected. It has been proven that infected farm animals are responsible for the infection of humans, being a source of the agent of the disease. The pathogens of leptospirosis are microorganisms of the genus *Leptospira*, belonging to the independent family, *Leptospiraceae,* of the order *Spirochaetales*. The genus, *Leptospira,* combines 64 species and more than 300 serovars. Leptospirosis in animals is only caused by two subclades: the pathogens (which includes 13 species) and the intermediates (11 species) [6,7,8]. More recently, animal leptospirosis has attracted increased attention due to the large number of outbreaks in humans worldwide [9]. WHO has identified leptospirosis as a tropical disease of global importance requiring further research to understand its epidemiology, ecology, and pathology [4,10].

The animals infected with *Leptospira* contaminate the environment via the excretion of bacteria through their urine, which represents an indirect route of leptospirosis transmission. A pathogen transmission occurs either directly through a susceptible animal’s contact with infected urine and other body fluids of an infected animal or via *Leptospira*-contaminated soil or water [11,12,13]. Historically, leptospirosis was known as an environment-borne infection, even before its etiological agent was identified [14,15,16] and soils in endemic areas were treated as its “environmental reservoirs” [17,18,19,20].

The incidence of leptospirosis varies in space and time and is closely related to climatic, ecological, and local socioeconomic factors [14,15,21]. A higher incidence is observed in tropical, humid, and temperate regions, especially during the rainy seasons, both in urban and rural areas [22,23], while research dedicated to leptospirosis patterns in northern regions are rare.

Based on the available data, it can be concluded that when assessing the transmission of leptospirosis to a susceptible livestock population, the geographical scale should be taken into account, in addition to global risk factors, in order to assess the local environmental and socioeconomic factors affecting animal infection [24]. Typically, massive outbreaks of leptospirosis in urban settings are associated with the abundance of rodents, being asymptomatic carriers of leprospires. In rural areas, outbreaks are linked to agricultural processes, such as animal breeding, while seasonal peaks exist in most affected areas suggesting that numerous environmental factors, such as torrential rains, floods, and cyclones may act as “triggers” of the disease outbreaks [25,26,27,28,29,30,31,32,33,34,35]. Environmental changes in human habitat conditions and urbanization combined with climate changes are the most important risk factors in the occurrence and spread of present and future leptospirosis outbreaks [18,36,37].

The complexity of pathogen transmission in leptospirosis is a serious problem for determining control strategies, especially in endemic areas. Epidemiological surveillance of the affected territories is the main and central step in the sequence of measures aimed at the rehabilitation of those territories after a leptospirosis outbreak [37,38,39]. Therefore, there is a need to select precise and cost-effective tools to improve existing surveillance procedures and strengthen control strategies. Geographic information systems (GIS) and geospatial statistics are now significantly improved and widely used in public health research. They can improve epidemiology and disease control measures. The goal of this study was the predictive modeling of the leptospirosis risk zones in the Republic of Sakha (Yakutia), Russian Federation (hereinafter RSY), in connection with a set of environmental and socioeconomic factors using the method of ecological niche modeling, and forecasting changes in risk levels due to the expected climate changes for the period up to 2060. To achieve this goal, the following objectives were formulated:To summarize the epidemiological data on the occurrence of animal leptospirosis in the RSY for 1995–2019;To identify the main environmental and socioeconomic factors contributing to the occurrence of animal leptospirosis outbreaks in the RSY;To develop an environmental suitability based leptospirosis risk map using the MaxEnt ecological niche model, subsequently averaging the risks within municipal areas to provide zoning in accordance with the currently applied veterinary practice.

## 2. Results

### 2.1. Epidemiological Analysis

From 1995 to 2019, 434 outbreaks of leptospirosis in animals were recorded in the RSY (Figure 1). In total, for this period, 2728 animals were infected, of which 31.4% were cattle (859 animals), and 57.2% were horses (1561 animals). The proportion of infected pigs and sheep was 7.6% and 1.2%, respectively. The rest of the infected animals were domestic (cats, dogs), and wild animals (reindeers, rabbits, etc.), which together sum up to 2.6% of all infected animals. The yearly incidence of leptospirosis in livestock species (cattle, horses, pigs, and sheep) in the RSY for the period from 1995 to 2019 is presented in Figure 2. Based on the data of recorded cases for this period, most outbreaks were noted in 2008. In total, there were three periods during which an increase in the incidence could be observed: (1) from 1995 to 2008; (2) from 2014 to 2016; and (3) from 2017 to 2018. The seasonality diagram suggests that a peak of the disease incidence was observed in the warm season (Figure 3). The incidence of leptospirosis reached its highest value in March and June, amounting to 15.9% and 15.4% of all outbreaks, respectively. In cattle, the maximum increase in incidence occurs in the spring–summer period: starting in March (12.2%) and April (17.8%). The peak incidence in cattle was in June (27.3%). In horses, the peak incidence was observed also in the spring–summer period, but the maximum incidence was found in both March (17.37%) and June (14.55%). In pigs, the seasonality pattern was different from that in cattle and horses. The incidence peaks in pigs were observed in January, February, April, June, and September. This is presumably related to the return of the Leptospira hosts: mice, voles, and rats in pig farms. The incidence of cattle and horses may be determined by the timing of the grazing period, which in the RSY normally lasts from June to September for cattle and almost year-round for horses.

### 2.2. Environmental Niche Modeling

An environmental niche model, implemented with MaxEnt, demonstrated a good ability to distinguish between true presence and pseudoabsence data (AUC = 0.930 ± 0.026). The most important environmental risk factors related to the leptospirosis presence points in the RSY were: mean temperature of the wettest quarter (Bio_8), altitude (alt), land cover, mean diurnal range (bio_2), temperature seasonality (bio_4), precipitation of the wettest month (bio_13), cattle density, and soil pH, and the importance of the permutations was 10.3%, 36.3%, 10.8%, 13.2%, 5.2%, 6.3% 5.0%, and 4.1%, respectively. The importance of these predictors was also confirmed by Jackknife analysis. The response curves, indicating the model gain using only a particular variable, are presented in Figure 4 for the eight most important variables (listed above). The curves suggest a maximum suitability to leptospirosis transmission in areas with the mean temperature of the wettest quarter being approximately 14 °C and above, low altitudes (less than 500 m above sea level), considerable variation in the monthly temperatures within a year (10 °C and above), pronounced seasonal temperature deviation of about ±20 °C, relatively low maximum precipitation of ~20–60 mm/month, relatively high cattle population density (more than 0.5 heads/km^2^), and soil pH of 5.5 and above. The land cover categories most closely associated with the risk of leptospirosis were: Urban and Built-Up areas, Open Deciduous Needleleaf Forest, Open ground and rock outcrops, Coastal vegetation, Permanent Wetlands, and Grasslands. The soil units corresponding to the most suitable areas were Histic Gleysols Dystric, Gleyic Albeluvisols Abruptic, Haplic Podzols, and Haplic Cambisols Eutric. The rest of the variables used for modeling did not demonstrate a significant contribution to the model gain, while still providing a reasonable influence on the probability of having a leptospirosis outbreak. Lower maximum green vegetation fraction values were more suitable, showing a monotonic decrease over the study range. The distance to the nearest water body appears to have a negative relation with leptospirosis locations, suggesting that most of the recorded cases occurred not farther than 2500 m from water.

Suitability maps for the current and future climate are shown in Figure 5a,b, respectively, while Figure 5c shows a map of statistically significant suitability changes in the future, as compared to the current conditions. The maps clearly demonstrate that the most suitable areas are present in the south-eastern part of the RSY, which is also the most inhabited territory of the Republic, and extend in the direction of major rivers.

### 2.3. Leptospirosis Risk Mapping

Gridded suitability maps were converted into categorical zoning maps based on the proportion of high-risk cells within each administrative district. The risk maps for the current climatic conditions (Figure 6a) and for the projected climate (Figure 6b) suggest the existence of high- and medium-risk zones in the south-eastern part of the RSY. In the current climate, the high-risk zone includes: Namsky, Ust-Aldansky, Tattinsky, Alekseevsky, Churapchinsky districts, and the city of Yakutsk. In addition, 14 districts can be rated as medium-risk areas. Under the projected climate, the Medino-Kangalassky, Amginsky, and Ust-May districts may fall into the high-risk zone, and the medium-risk zone will increase by three districts, providing an expansion of leptospirosis into nearly half of the RSY territory.

## 3. Discussion

In this study, for the first time, we collected data on leptospirosis outbreaks in the subarctic region of Russia from 1995 to 2019. We identified a number of environmental and socioeconomic factors that are associated with the disease’s geographical spread. We used the ecological niche modeling (ENM) approach to better understand the relationship between the distribution of leptospirosis outbreaks and environmental variables and to construct predictive maps that illustrate the RSY territory suitability to the leptospirosis emergence.

The results of many studies have shown the importance of accounting for environmental factors when studying the epidemiology of leptospirosis [21,40,41].

Environmental factors play a significant role in the emergence of natural focal diseases, directly affecting pathogenic microorganisms or affecting the distribution and number of hosts and carriers of diseases and forming favorable conditions for the persistence and transmission of the disease [21]. Since the ENM tools were first used to predict outbreaks of leptospirosis in the subarctic region of Russia, we tested all commonly used environmental variables as potential predictors. While not directly comparable with the results of other authors due to the use of different variables, our findings reveal very similar distribution patterns of recorded leptospirosis outbreaks. Our choice of final variables was ultimately determined by the procedure of reducing multicollinearity, potentially leaving overboard some more general parameters, e.g., the yearly mean temperature and precipitation. Our results rely on more specific parameters, the interpretation of which may be difficult to compare with the results of other studies. Nevertheless, the revealed patterns may be treated as very similar to the ones reported in [21,40,41]. The mean temperature of the wettest quarter, temperature seasonality, temperature variation, and soil pH were the most important environmental factors determining the geographical spread of leptospirosis in the Republic of Sakha (Yakutia). In general, the combination of these factors suggests that the persistence and transmission of *Leptospira* favor humid areas with a considerable temperature variation throughout the year. Since leptospirosis is an infectious disease that can be transmitted through water, we used the Euclidean distance to nearest water sources for the transmission of leptospirosis indirectly. Unfortunately, the corresponding variable does not significantly affect the geographical spread of leptospirosis, while demonstrating an expected pattern of the suitability decrease as the distance to water increases. From a socioeconomical perspective, only the cattle population density was used, which was found to be statistically significantly associated with the recorded outbreaks. This factor may be treated as a proxy of human presence and economic activity in the sparsely populated area of the RSY, as well as an indicator of the cattle population’s exposure to leptospires.

There is a lot of evidence that virulent *Leptospira* can survive and remain infectious in an environment for several months, particularly in soils [42,43]. According to studies of other authors, leptospires are more often isolated from soils, rather than from freshwater samples [44,45,46]. These data confirm the hypothesis that soils can be a reservoir of *Leptospira* spp. and specifically of its pathogenic strains [47,48]. Due to the climatic conditions, the soils in the RSY are in the frozen state for most of the year. This may explain the fact that most of the RSY has a low leptospirosis suitability, while an elevated suitability was found in proximity to rivers and populated places that occupy the most naturally favorable areas of the Republic. The soil units that demonstrated most significant associations with leptospirosis outbreaks can be described as tundra and taiga-specific soils, with a relatively high organic content. Soil pH was found to more significantly influence suitability to leptospirosis, with higher values being more preferable, which correspond well to the findings in [40,41].

The combination of low altitudes and specific land cover categories associated with the known locations of leptospirosis outbreaks may either suggest a data bias, related to the georeferencing of outbreaks to the nearest settlement, or point to the disease emergence in anthropogenic landscapes, where the transmission of the pathogen to farm animals and humans is possible. To reduce a potential bias of data towards the populated areas, we introduced a “data bias correction factor” to the MaxEnt model, which was the density of settlements. Such a factor assumes that the input presence locations might be primarily recorded in close proximity to populated places and corrects the resulted suitability distribution accordingly. The observed pattern of the suitability response to the maximum vegetation fraction with lower maximum green vegetation fraction (MGVF) values being more suitable may also suggest a primary disease transmission within inhabited areas, which are less covered by vegetation.

The predicted distribution of the suitability change in a future climate suggests an increase of leptospirosis risk over the large area of the RSY. Some areas in the north and northeast of the Republic, however, demonstrate a statistically significant reduction of suitability. The observed pattern looks quite similar to the estimated change of the Bio_8 variable that stays for the Mean Temperature of the Wettest Quarter. Considering an expected temperature rise over the entire territory of the RSY, one can assume a wettest season’s shift towards colder months that will lead to the decrease of the mean temperature. Those areas of expected suitability reduction mainly correspond to lowlands surrounded by mountainous areas, such as Verkhoyansk and Chersky Ranges. Such territories are characterized by the stagnation of cold air masses flown down from mountains and thus normally demonstrate relatively lower air temperatures than flat or highland areas.

The results obtained allow optimizing the long-term control and monitoring program aimed at leptospirosis prevention in high-risk areas that, in particular, includes compulsory vaccination of domestic animals.

## 4. Materials and Methods

### 4.1. Study Area

The Republic of Sakha (Yakutia) (RSY) is one of the 85 federal subjects of the Russian Federation, located in its north-eastern part. Its mainland extends from 55° to 74° North and from 105° to 163° East (Figure 1). With an area of more than 3,000,000 km^2^, the Republic is the largest federal subject of the Russian Federation, as well as the largest administrative unit in the world. In addition, the population density is only about 0.3 persons/km^2^, which is one of the lowest rates in the Russian Federation and in the world. More than one-third of the entire territory of the Republic is located beyond the Arctic Circle, more than 80% of the area is covered by forests, and more than two-thirds are mountains and plateaus. The Republic has a large number of natural water bodies (rivers and lakes). The climate ranges from continental in southern and central parts to subarctic and arctic in the north. The Republic is rich in minerals (coal, natural gas, diamonds, gold, etc.). The main inhabited zones are located along the major rivers, as well as around large subsoil deposit locations. The territory of the Republic is divided into 36 districts (“uluses”), which are second-level administrative units. The agricultural activity is mainly cattle farming, horse breeding, and reindeer husbandry.

### 4.2. Leptospirosis Data

The leptospirosis data were obtained from two official sources. The regional veterinary service reports provided information on the leptospirosis cases in farm animals. Such data are based on farm owners’ claims followed by laboratory confirmation. Additionally, statistical veterinary reports of the Federal Service for Veterinary and Phytosanitary Surveillance (Rosselkhoznadzor) were used that contain the results on the positive testing of both farm and wild animals during the passive surveillance program being implemented in the RSY. The surveillance program implies testing of 25% of the susceptible farm animals with occasional testing of wild animals. According to the currently effective National Standard “GOST 25386-91”, the laboratory confirmation was carried out by the Yakut Republican Veterinary Testing Laboratory via a microagglutination reaction test (MAT) using a set of 7 reference cultures: Pomona, Tarassovi, Canicola, Hebdomadis, Sejroe, Grippotyphosa, and Icterohaemorrhagiae. The reaction was evaluated in a positive cutoff, with a serum dilution of 1:50 for unvaccinated, and 1:100 for vaccinated animals with dark-field microscopy. For previously vaccinated animals, testing was conducted at least 3 months after the vaccination. In case of a positive MAT result and no clinical signs, confirmatory testing by real-time PCR was used.

In total, 454 outbreaks of leptospirosis were recorded in the RSY during the study period of 1995–2019 (Figure 1). In this study, we defined an outbreak as the reported occurrence of leptospirosis in a geographically localized animal population (herd on a certain pasture, farm, village, etc.) with at least one infected or seropositive animal, confirmed by laboratory research methods. The database contains outbreaks of leptospirosis detected in cattle, small ruminants, pigs, horses, dogs and cats, and wild animals, including rabbits, rats, and mice.

The following indicators, important for further modeling, were recorded for each outbreak: Geographical coordinates, exposed animal species, and their number, as well as the date the disease, were diagnosed. After excluding inaccurate records (in particular, the cases without geographical coordinates or with incorrect ones), the database contained 434 outbreaks of leptospirosis, in which 2728 out of the 46,569 susceptible animals were exposed. Taking into account the number of repeated outbreaks recorded at different time periods in one location, the total number of unique locations was 207. The map of outbreaks is shown in Figure 1. The geographic data are converted into a shapefile format for visualization and further modeling.

### 4.3. Modeling Approach

We used the method of ecological niche modeling with the principle of maximum entropy (MaxEnt) to identify the relationship of recorded outbreaks of leptospirosis with a complex of abiotic and biotic variables. This approach, described by Phillips [49], is currently one of the most popular methods for modeling the distribution of a studied phenomenon in relation to environmental factors based on presence-only data [50,51].

The principle underlying the maximum entropy method is to search for the probability distribution of each factor that would most uniformly describe its change within the range specified by the presence data (i.e., would have a maximum informational entropy) [50]. The resulting map shows the probability that a combination of environmental factors at each cell of the study area is suitable for the studied phenomenon. Predictive power is estimated by the ability of the model to distinguish between the presence and pseudoabsence data and is expressed by the area under the ROC curve (AUC). This indicator shows the probability that a randomly selected presence point will have a higher prognostic value than a randomly selected pseudoabsence point [52]. In addition, this method makes it possible to evaluate the contribution of each variable to the obtained model and to draw a conclusion about the most significant biological factors by means of the Jackknife technique, which is based on the sequential exclusion of each variable from the model and a comparison of the model gain with and without the variable, as well as with the variable only.

MaxEnt modeling was performed using cross-validation, with the presence data divided into 10 folds and each fold in turn used for testing. Summary results, thus, include average values over 10 replications as well as standard deviation limits. Within each replication, 5000 iterations were used to reach the highest gain with 0.00001 convergence threshold.

Using two sets of BIO variables (see Section 4.4.1 below), modeling was performed both for the current climatic conditions and for a projected climate for the period 2041–2060.

To account for a potential data bias related to a sampling distribution shifted towards populated places, we included a “data bias” parameter expressed as the settlement density. It introduces an assumption that incidence records are more likely to be made in or in close proximity to towns, settlements, and villages.

A comparison of the suitability for the current and projected climates was conducted by mapping statistically significant estimated changes for each cell of the study area. The map is developed by subtracting the current suitability value from the predicted suitability and then normalizing the difference with the standard deviation calculated for the current suitability. Three classes of expected changes were distinguished:Difference within ±1 of the standard deviation was recognized as statistically insignificant;Difference below −1 of the standard deviation indicates the places in which the suitability reduction was predicted;Difference above +1 of the standard deviation indicates the places in which an increase in suitability was predicted.

To compare risks at the level of administrative divisions, zoning was performed by calculating the proportion of spatial cells with a suitability above 0.5 (“high-risk cells”) for each municipal district, both for the current and projected climates, and, subsequently, categorizing the risk levels in three classes based on the distribution of the obtained proportions [53,54].

### 4.4. Explanatory Variables

#### 4.4.1. Climatic Variables

We used a set of bioclimatic variables BIO as climatic factors for the current and projected climates (https://worldclim.org/data/index.html). This set consists of 19 variables (BIO_1 … BIO_19), describing the peak and trend values based on the average monthly air temperatures and precipitation: (I) for 1970–2000, according to weather stations data; and (II) projected for the period 2041–2060, calculated on the basis of the downscaled INMCM4 climate model. This is the atmosphere-ocean global climate model developed by the Institute of Numerical Mathematics, Russian Academy of Science. The model has participated in the International Coupled Model Intercomparison Project, phase 5 (CMIP5) [55]. A modeling scenario was used that corresponds to a representative concentration pathway of RCP8.5. This scenario determines that the concentration of greenhouse gas and anthropogenic effects in the atmosphere raises in the toughest trajectory and presents the most stringent option for climate change. The initial spatial resolution of the data is 30 arc seconds.

#### 4.4.2. Animal Host Density

Leptospirosis is detected mainly in those areas where people interact with animals, with excretions of infected animals or with contaminated environmental objects [11]. Rodents and some domestic animals, such as pigs and dogs, are considered the riskiest for the transmission of leptospires to humans [31]. Leptospirosis is considered an epidemic infection that can be transmitted directly to humans from water, soil, and urine-contaminated food from infected host animals [12]. We used the density of the cattle population as the main species of farm livestock in the study region. This indicator was used as a proxy for the intensity of animal husbandry and human agricultural activity, increasing the likelihood of contact between animals and wild species. Animal density data were obtained from the FAO geoportal (http://www.fao.org/geonetwork/srv/en/main.home) in the raster format.

#### 4.4.3. Proximity to Water Bodies

The literature sources confirm that the pathogenic Leptospira is associated with the presence of water bodies, including rivers, streams, lakes, and springs. Therefore, the distance to natural and artificial water sources is a potential risk factor for leptospirosis [56].

The database of freshwater reservoirs (rivers, lakes) in the RSY was recovered from the vector dataset, “Digital model of Russia on a scale of 1:500,000” (https://www.esri-cis.ru/products/). The distance to the nearest water body was calculated using the Euclidean distance tool (Spatial Analyst, ArcMap).

#### 4.4.4. Land Cover Type

As a source of data on the types of land cover in the Russian Federation, we used a digital map, created according to the Proba-V satellite system for the years 2000–2018, with an original spatial resolution of 100 × 100 m [57]. The categories of land cover presented within the entire database are listed in Table A1.

#### 4.4.5. Vegetation Index

An average maximum green vegetation fraction (MGVF) for 2001–2012 was used as an indicator of the intensity of vegetation within the model area [58].

#### 4.4.6. Soil Type

Data on the soil type in the model territory were obtained from the Unified State Register of Soil Resources of Russia (http://egrpr.esoil.ru/content/1DB.html), which has 255 soil units, in accordance with the world harmonized database of soil data (http://www.fao.org/soils-portal/soil-survey/soil-maps-and-databases/harmonized-world-soil-database-v12/en/). The soil units presented in the RSY are listed in Table A2. The original vector data were converted into a raster format.

#### 4.4.7. Soil pH

To study the potential relationship of leptospirosis outbreaks with soil acidity, a variable was included in the model that displays the soil pH at zero depth, according to the ISRIC-World Soil Information. The data are presented in raster format, with an initial resolution of 250 × 250 m^2^ [59].

### 4.5. Data Processing and Software

The Asia Lambert Conformal Conic map projection was used for mapping and geospatial analysis. All geospatial variables presented in raster format were reduced to a single spatial resolution of 1 × 1 km and converted to the ASCII format. A preliminary correlation analysis of all variables was performed to reduce the multicollinearity of the MaxEnt model using the “usdm” package in the R 3.6.2 software environment [60]. At the first step, of each pair of highly correlated variables (Pearson correlation coefficient above 0.85) the one was excluded with a larger VIF (Variance Inflation Factor). For the remaining variables, VIF was analyzed, and indicators with VIF > 10 were additionally excluded. The final set of predictors is presented in Table 1. Statistical data processing was carried out using the MS Office Excel package (Microsoft, Redmond, WA, USA). To process the geodata and create maps, the ArcMap 10.7.1 geographic information system (ESRI, Redlands, CA, USA) was used. Ecological modeling with the maximum entropy method was carried out using the MaxEnt software [61].

## 5. Conclusions

This article discusses, for the first time, the problem of the spread of leptospirosis in the Republic of Sakha (Yakutia), a subarctic region with a continental to arctic type of climate.

The results of this study demonstrate that the distribution of leptospirosis cases in the Republic is significantly influenced by complex landscape and climatic factors, which corresponds well to the conclusions made by other authors. The findings suggest an expansion of the leptospirosis risk zone in the Republic under the projected climate change in 2041–2060. The method of risk zoning may be recommended to the national veterinary service as a basis for conducting targeted surveillance of leptospirosis in the environment and in livestock populations.

## Figures and Tables

**Figure 1 pathogens-09-00504-f001:**
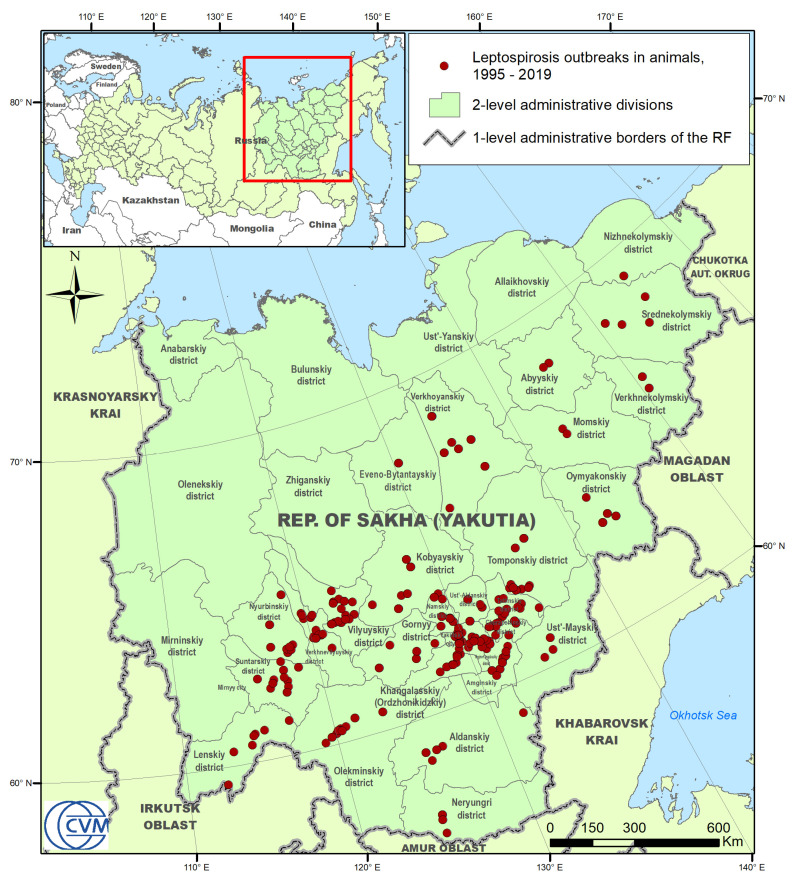
The distribution of recorded leptospirosis outbreaks in animals in the Republic of Sakha (Yacutia), 1995–2019.

**Figure 2 pathogens-09-00504-f002:**
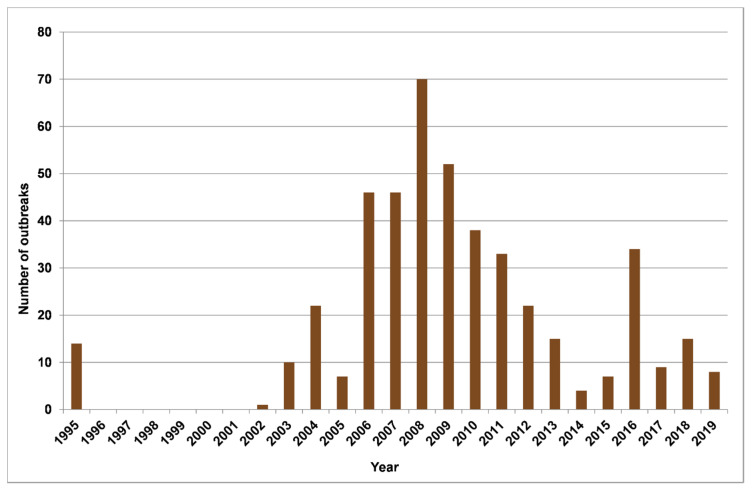
Yearly incidence of leptospirosis in livestock and wild animals in the Republic of Sakha (Yacutia), 1995–2019.

**Figure 3 pathogens-09-00504-f003:**
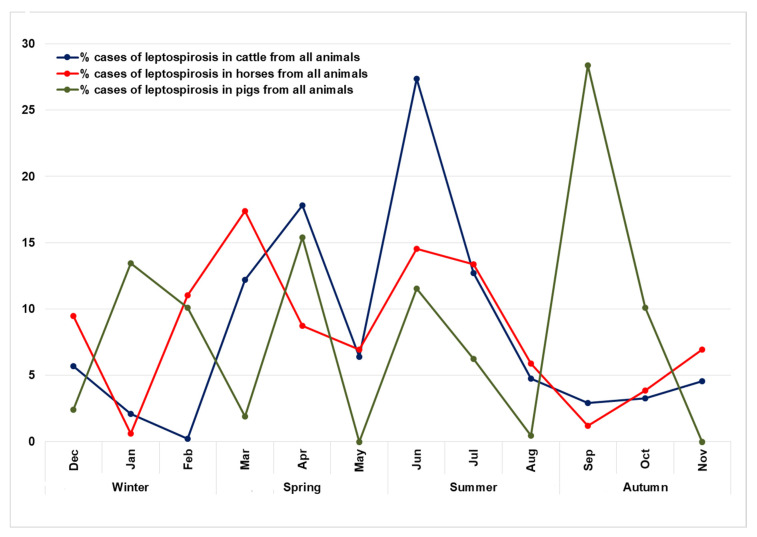
Seasonality of leptospirosis outbreaks in cattle, horses and pigs in the Republic of Sakha (Yacutia), 1995–2019.

**Figure 4 pathogens-09-00504-f004:**
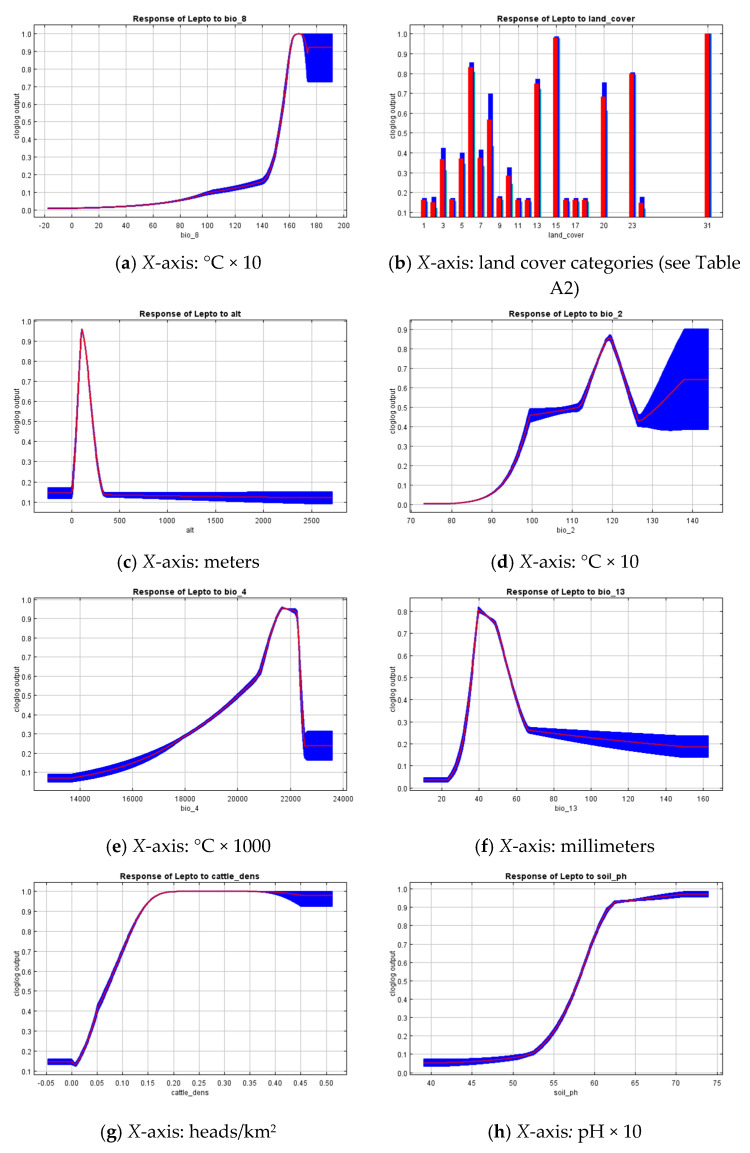
Response curves for 8 variables that contribute the most: (**a**) Bio_8—Mean Temperature of Wettest Quarter; (**b**) land cover type; (**c**) altitude; (**d**) bio_2—Mean Diurnal Range; (**e**) bio_4—Temperature Seasonality; (**f**) bio_13—Precipitation of Wettest Month; (**g**) cattle density; (**h**) soil pH. Red lines indicate average trends over 10 model replications while blue areas denote the standard deviation limits. The *Y*-axis is relative suitability calculated only using the particular variable. The *X*-axis variables’ units are presented at a footnote of each diagram as well as listed in Table 1.

**Figure 5 pathogens-09-00504-f005:**
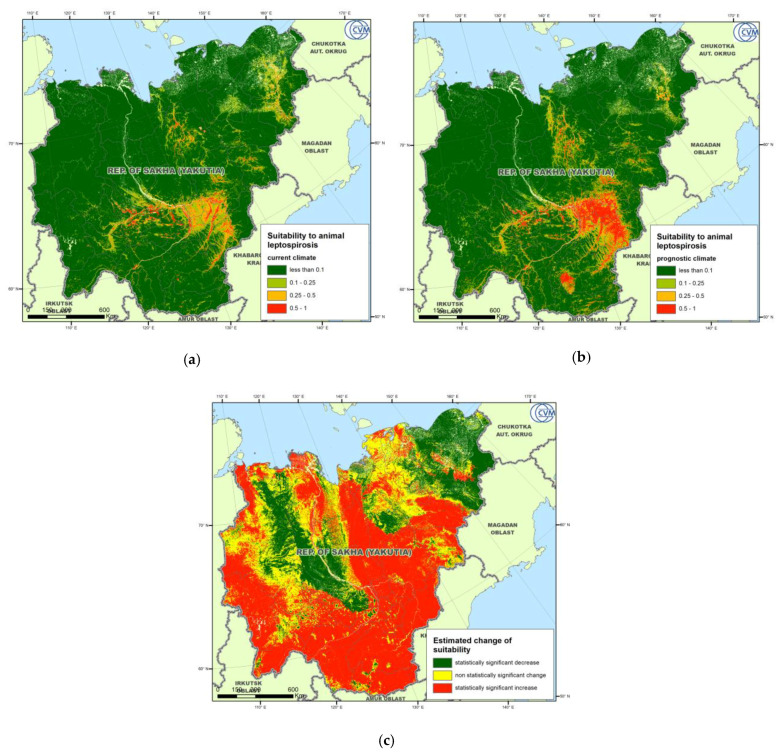
MaxEnt-derived gridded suitability maps for the current climate (**a**), projected climate (**b**), and statistically significant changes (**c**).

**Figure 6 pathogens-09-00504-f006:**
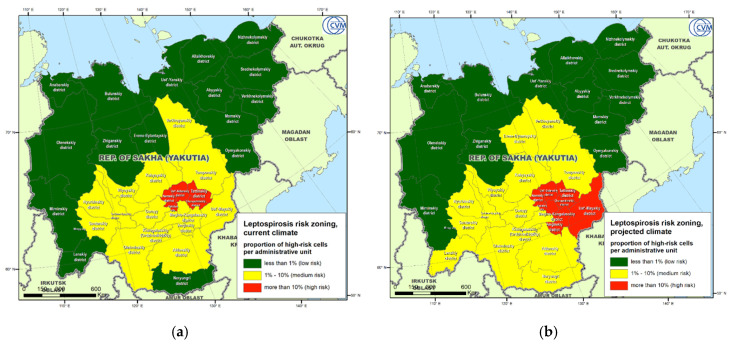
Republic of Sakha (Yakutia) Leptospirosis risk zoning for the current climate (**a**) and for the projected climate (2041–2060) (**b**).

**Table 1 pathogens-09-00504-t001:** Environmental and socioeconomic factors used as explanatory variables in an ecological niche model after removing correlated variables and variables with a high VIF.

Variable Name	Variable Description	Units	Data Type
Alt	Altitude	Meters	
Bio_2	Mean Diurnal Range (Mean of monthly (max temp–min temp))	°C × 10	Continuous
Bio_4	Temperature Seasonality (standard deviation × 100)	°C × 1000	Continuous
Bio_8	Mean Temperature of Wettest Quarter	°C × 10	Continuous
Bio_9	Mean Temperature of Driest Quarter	°C × 10	Continuous
Bio_13	Precipitation of Wettest Month	Millimeters	Continuous
Bio_14	Precipitation of Driest Month	Millimeters	Continuous
Bio_15	Precipitation Seasonality (Coefficient of Variation)	Proportion	Continuous
MGVF	Maximum Green Vegetation Fraction	Proportion	Continuous
Land cover	Land cover type	Land cover categories (see Table A1)	Categorical
Soils	Soil type	Soil categories (see Table A2)	Categorical
Soil pH	Soil pH at zero depth	pH × 10	Continuous
Water distance	Euclidean distance to the nearest freshwater body	Meters	Continuous
Cattle density	Density of cattle	Heads/km^2^	Continuous

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
