# Peer review of "Environmental Risk of Leptospirosis in Animals: The Case of the Republic of Sakha (Yakutia), Russian Federation"

_pathogens, 2020, doi:10.3390/pathogens9060504_

Round 1

Reviewer 1 Report

This paper reports on an ecological niche model to estimate and predict the risk of Leptospira infection in animals in a Russian region using several ecological and climatic variables.
Overall the paper is well written, very interesting both for the topic and the methodology. Apologies to the authors for taking so much time to review the paper that I really enjoyed reading.
My comments are mostly on the need for more details, especially around the methodology and source of data.

General comments:
Throughout the manuscript the authors mix up leptospirosis, Leptospira infection and Leptospira exposure. Leptospirosis refers to the disease itself, in animals with clinical signs. Exposure can be present or past, and is what serology detects. This needs to be clarified throughout the manuscript. See also my specific comments on the methods.
For example in the abstract l. 16 the authors say "exposure to the disease" or l 46. "Leptospirosis infection", which is not possible; l. 37 not only sick animals can infect humans, asymptomatic infected animals can also be a (and most likely are the main) source of human infection.

Abstract:
I undertand the authors have been limited by the word count. It would be great to see some more precisions here, such as the most important environmental factors and the value of the AUC to support the claim of "very good performance".

Introduction
Generally the introduction is a bit confusing as the authors navigate between animal and human information without always specifying which one they are talking about.
l 31. I think the general claim that leptospirosis as a whole is natural focal is debatable. This is true for some hosts-serovars associations, but for adapted serovars in their reservoir hosts the maintenance can be done independently of environmental conditions, e.g. Hardjo in cattle. This is possibly true for the serovars and conditions in Russia but it would need to be supported by reference(s).
l 40. There are many more species of Leptospira, see for example Guglielmini et al. PLoS NTD 2019 Genus-wide Leptospira core genome multilocus sequence typing for strain taxonomy and global surveillance, or Thibeaux et al. Microb Genom 2018 Deciphering the unexplored Leptospira diversity from soils uncovers genomic evolution to virulence.
l 41-42. it may be worth to precise you are talking about outbreaks in humans, not animals
l 47-48. Not only, transmission also occurs via direct or indirect (not only soil and water) contact with urine of shedding animals.
l 58-59. The paragraph seems to be about livestock leptospirosis but I doubt outbreaks occur in urban settings, and the references are about human leptospirosis. This sentence is possibly out of place.
l 61. What are these "environmental triggers"? or at least give some examples.
l 81. Risk map may be a better word, as prognostic of disease has a different meaning compared to climate. Also is the objective really to develop risk maps of leptospirosis outbreaks, so about clinical expression and not endemic presence?

Results
l 93. This is an incidence, not an incidence rate.
l 94. The 3 periods mentioned are not reflected on Figure 2. I see a peak in 2008, one in 2016 and possibly another one in 2018.
Figure 3. Please add the seasons in the figure, and specify what the % in the legend refers to: is it the % of all cattle cases that were recorded in a given month, or for a given month the % of all cases that were cattle?
l 117. Please restate here what is predicted by the model: demonstrated a good predictive ability of XXX.
l 121-123. There are 2 values for the contribution of cattle density, 5% and 7.7%.
Figure 4: for all plots of continuous variables, please state the unit in the x axis instad of just Table 1 to facilitate reading; rename the y axis so it can be self explicit; for the barcharts please use error bars instead of overlapping bars that are difficult to see.
Figure 6: Please indicate the risk intervals corresponding to the different categories.

Discussion
Again please make sure the terminology on disease vs. infection vs. exposure is clear and coherent with the data used to build the model.
l 192. leptospirosis is a disease that CAN BE transmitted through water would be better.
Please develop at the end of the discussion the relevance of the findings for control, prevention and surveillance of leptospirosis in the region.
Depending on the precisions that need to be added in the methods (See below), it may be useful to discuss the risk of selection bias linked with the outbreak recording process.

Methods
l 239. What does "severely continental" mean?
l 245-247. Please explain where the data in the 2 sources used (regional veterinary service reports and statistical veterinary reports) and in what they differ. Could some data have been reported in both? Which animals were tested? Only those with clinical signs? Animals with clinical signs + in contact animals?
l 249-250. Please detail the definition of an outbreak. Was it at least one animal seropositive in a given population?
l 252-253. Please list the serovars used in the MAT panel and what cutoff what used for positivity.
l 254. Were the rabbits, rats and mice domestic or wild? Also the presence of wild animals means that "animals" or a similar term should be used instead of "stock" in the rest of the manuscript.
l 256. If only MAT was used, the data is on exposed animals, not infected, as MAT correlates poorly with infection.
l 257. How many records were present in total and how many were excluded?
l 260. How was the difference made between repeated outbreaks and one outbreak prolonged over time?
l 291. Please replace "suitability levels" by "probability of suitability".
l 293. Please replace "prognostic value" by "predicted probability".
l 305 onwards. The date range used for the explanatory variables does not match the date range for the animal cases data, sometimes with no overlap at all. Please use the same date range, or if not possible please indicate why and discuss the potential effect on the model.
l 307. Please define BIOCLIM
l 310. Future climate data are usually called "projections", not "predictions", please correct in the manuscript.
l 310. Please detail the use of the INMCM4 model. Is it a local model, or a globnal model downscaled? What was the spatial resolution of the projections?
l 312. RCP8.5 is a "worst case scenario" that has been deemed implausible. Please use RCP 7 or 6 instead.
l 361-362. If 2 variables were found to have a correlation >0.85 were both of them removed? If only one, how did you select the one to remain in the model?
Table 1. Please edit the caption to make it more explicit. Something like "Environmental and socioeconomic factors used as explanatory variables in an ecological niche model after removing correlated variables and variables with a high VIF".

Reviewer 2 Report

The MS "Environmental risk of leptospirosis in animals: the case of the Republic of Sakha (Yakutia), Russian Federation" by Zakharova and co-authors provides an original study of spatial modelling of animal leptospirosis in Yakutia, Russian Federation.

The work is sound and mostly well presented.

The most original component of this study is that the region is largely beyond the Artic Circle, a very unusual climatic area for leptospirosis.

There are several issues that need to be addressed in the MS as it currently stands.

First, there is a massive problem with reference numberingand many references are incorrect and do not correspond to the citation and the numbers 62 and 63 cited in the text are not references. I trust there has been a significant mix in references (e.g. 36 is cited in text as Phillips, but ref 36 is not Phillips, etc...)

The background information on leptospirosis provided in the introduction is incorrect:

  • The global burden and mortality model cited (Costa et al 2015) evaluates the global mortality to be 58,900, not above 60,000.
  • the Leptospira the old taxonomy with only L. interrogans and L. biflexa is obsolete and should not be used anymore. The current number of Leptospira species is 64, not 2! https://www.ncbi.nlm.nih.gov/pubmed/31120895
  • The Order name is Spirochaetales
  • WHO unfortunately does not recognize leptospirosis as a neglected disease (though it could largely be)

Minor comments in the introduction are about

  • The unusual expression "natural-focal", that I could understand but is not very explicit
  • The wording "altitude above sea level" is redundant, I would prefer "altitude" alone or "elevation above sea level". This also applied elsewhere in the MS (e.g. line 119).

In the results section, I would suggest to give some background information on the farming techniques used, notably the months where animals (cattle, horses, pigs) live outside and the months when they are inside in barns (lines 96-105).

Maybe I have missed some information, but I could not understand how the leptospirosis risk is somehow normalized to density (the higher the number of animals, the higher the probability of an outbreak)? I sugget to make this clearer and notably include this in the results section, especially because he M&M only come after the results.

The figure 3 on seasonality is original, but not very easy to interpret. Although acceptable, I think it could be improved.

While reading the results (line 144-146), I was wondering about a possible bias in reporting. I appreciated the attention paid to this issue, but could only know that very late in the MS (line 287), so I suggest to indicate this both in the results and in the discussion (not only in the M&M section).

The use of "Bio_8" etc... in figure 4 is not very explicit. I would suggest to correct the figure panels or at least give the detailed meaning in the legend.

The wording "prognostic climate" is unusual and I suggest to change t for "future climate projections" (lines 161, 286, 291, 303, 307, 378).

I am questioning the usefulness of including both figures 5 and 6, I feel they somehow convey the same information. If so, one could be included in the appendix. If not, this should be made clearer to the reader.

The use of the wording Leptospiras (italicized plural) is incorrect. Please use either the common name (leptospires) or the latin name Leptospira (line 190 & 199).

I have three minor concerns too:

  • line 220, I suggest to reword "over the most territory of the RSY"
  • line 314, I suggest to translate the spatial resolution in meters in addition to arc seconds
  • line 338, the spatial resolution is stated to be 100x100m². Is it 100 m² (10x10m) or 100 x 100m (10,000m²)?

Round 2

Reviewer 1 Report

The authors have now addressed most of my comments.

I have 2 points remaining that need clarification:
- the authors state that the objective is to describe leptospirosis outbreaks, identify risk factors and create risk maps. The study however relies only on MAT data, which correlates only poorly with infection, and even less with leptospirosis, which is the clinical manifestation of Leptospira infection.
Please either modify the objectives and discussion accordingly or explain in the methods section how clinical manifestations were confirmed.

- the data sources haven't been explaiend. What process needs to happen for an animal to end up on either report? It looks like a combination of veterinary diagnoses and surveillance data. Does it need to have clinical signs, then the owner calls the vet, the vet investigates then report? what about wild animals then? What is the surveillance programme? for which species? is it active or passive? this information is crucial and must be included in the methods for the readers to assess the risk of selection bias and internal and external validity of the paper.

Reviewer 2 Report

The comments I have raised have been properly addressed.

I am sorry I did not pick this earlier, but I trust the word zoning (in legend boxes of figures 5 & 6) spells with a single N and should be corrected.

Some figures could be placed side by side during page setting, notably figures 6 a & b.

All panels of figure 4 should be placed onto the same page.
